

# Improved YOLOv4-tiny based on attention mechanism for skin detection

Ping Li[1,2], Taiyu Han[1], Yifei Ren[1], Peng Xu[1] and Hongliu Yu[1]

[1] Institute of Rehabilitation Engineering and Technology, University of Shanghai for Science and Technology, Shanghai, China

[2] Department of Biomedical Engineering, Changzhi Medical College, Changzhi, Shanxi, China

## ABSTRACT

**Background:** An automatic bathing robot needs to identify the area to be bathed in order to perform visually-guided bathing tasks. Skin detection is the first step. The deep convolutional neural network (CNN)-based object detection algorithm shows excellent robustness to light and environmental changes when performing skin detection. The one-stage object detection algorithm has good real-time performance, and is widely used in practical projects.

**Methods:** In our previous work, we performed skin detection using Faster R-CNN (ResNet50 as backbone), Faster R-CNN (MobileNetV2 as backbone), YOLOv3 (DarkNet53 as backbone), YOLOv4 (CSPDarknet53 as backbone), and CenterNet (Hourglass as backbone), and found that YOLOv4 had the best performance. In this study, we considered the convenience of practical deployment and used the lightweight version of YOLOv4, *i.e.*, YOLOv4-tiny, for skin detection. Additionally, we added three kinds of attention mechanisms to strengthen feature extraction: SE, ECA, and CBAM. We added the attention module to the two feature layers of the backbone output. In the enhanced feature extraction network part, we applied the attention module to the up-sampled features. For full comparison, we used other lightweight methods that use MobileNetV1, MobileNetV2, and MobileNetV3 as the backbone of YOLOv4. We established a comprehensive evaluation index to evaluate the performance of the models that mainly reflected the balance between model size and mAP.

**Results:** The experimental results revealed that the weight file of YOLOv4-tiny without attention mechanisms was reduced to 9.2% of YOLOv4, but the mAP maintained 67.3% of YOLOv4. YOLOv4-tiny's performance improved after combining the CBAM and ECA modules, but the addition of SE deteriorated the performance of YOLOv4-tiny. MobileNetVX_YOLOv4 (X = 1, 2, 3), which used MobileNetV1, MobileNetV2, and MobileNetV3 as the backbone of YOLOv4, showed higher mAP than YOLOv4-tiny series (including YOLOv4-tiny and three improved YOLOv4-tiny based on the attention mechanism) but had a larger weight file. The network performance was evaluated using the comprehensive evaluation index. The model, which integrates the CBAM attention mechanism and YOLOv4-tiny, achieved a good balance between model size and detection accuracy.

Corresponding author
Hongliu Yu, yhl_usst@outlook.com

## INTRODUCTION

A convolutional neural network (CNN) is a machine learning model in a supervised learning framework. In 2012, AlexNet first used CNN for image classification (*Krizhevsky, Sutskever & Hinton, 2017*), winning the ImageNet large scale visual recognition challenge by an overwhelming margin. Since then, CNN has been widely used in computer vision tasks such as image classification (*Liu, Soh & Lorang, 2021*) and object detection (*Zhou et al., 2022*). By using massive data as learning samples, we can obtain a CNN model with analysis, feature representation, and recognition capabilities in order to achieve skin detection.

Skin detection is a prerequisite for bathing by automatic bathing robots. The intelligent bathing system detects human skin in the bathing environment based on vision sensors. Skin detection in bathing scenes is a challenging task. The bathing environment is full of water mist and various lighting and backgrounds. A skin detection algorithm generally extracts skin features and then classifies them using a classifier. Traditional skin detection typically exploits handcrafted features to distinguish between skin and non-skin zones, such as color, texture, and statistical features. Handcrafted features are not sensitive to environmental changes and are insufficient for bathing scenarios. Skin detection based on machine learning, which generally uses supervised methods to construct detectors in order to extract skin features, is less influenced by environmental factors and has gained more applications in recent years. *Salah, Othmani & Kherallah (2022)* utilized CNN trained by skin and non-skin patches to detect skin pixels. *Kim, Hwang & Cho (2017)* exploited two CNNs for skin detection and compared performance using different training strategies. *Lin et al. (2021)* conducted CNN-based facial skin detection and optimized the CNN using the Taguchi method.

Instead of merely identifying skin and non-skin areas, we needed to provide the robot with information about specific skin areas (hands, feet, trunk, *etc.*) to clean up skin using different modes. We faced a multi-classification problem rather than a secondary classification problem. In application areas, one-stage object detection models based on CNN achieve good real-time performance and are computationally efficient. YOLO series are typical one-stage algorithms. YOLOv2, YOLOv3, YOLOv4, YOLOv5, and YOLOv7 are anchor-based algorithms that use anchors as the prior knowledge of the bounding box. YOLOv2 is not good at detecting small targets and uses Darknet19 as the backbone (*Redmon & Farhadi, 2017*). YOLOv3 adopts Darknet53 to extract features (*Redmon & Farhadi, 2018*). YOLOv4 uses CSPDarknet53 to extract features and uses SPP and PANet for feature fusion (*Bochkovskiy, Wang & Liao, 2020*). The backbone and the neck parts of YOLOv5 include the CSP structure, and the Focus structure is proposed. Four models (YOLOv5s, YOLOv5m, YOLOv5l, and YOLOv5x) are provided. They have different depths and widths. The accuracy continuously improves, but the speed consumption also increases (*Xie, Lin & Liu, 2022*). YOLOv7 proposed the ELAN structure (*Wang, Bochkovskiy & Liao, 2022*). YOLOv1, YOLOv6, and YOLOX are anchor-free algorithms that lack the prior information of the bounding box and have better scene generalization in theory. YOLOv1 is less effective for small and dense targets (*Redmon et al., 2016*). YOLOv6

adopts SPPF and Rep-PAN structures, whose backbone is mainly composed of RepVGGBlock modules (*Li et al., 2022*). YOLOX includes standard versions (YOLOX-s, YLOLX-m, YOLOX-l, YOLOX-x, and YOLOX-Darknet53) and lightweight versions (YOLOX-Nano and YOLOX-Tiny) (*Ge et al., 2021*). YOLOX and YOLOv6 decouple the regression and classification in the detection head.

Our research is based on previous work by our team (*Li et al., 2021*) that found that YOLOv4 had a high mAP for skin detection in bath environments. The extensive computation led YOLOv4 having a slow speed after being deployed to embedded devices. In the study, we adopted YOLOv4-tiny (*Zhao et al., 2022a*), the lightweight model of YOLOv4, for skin detection and investigated the effect of attention mechanisms on YOLOv4-tiny. For full comparison, we also adopt other lightweight methods, namely, using MobileNetV1 (*Howard et al., 2017*), MobileNetV2 (*Sandler et al., 2018*), and MobileNetV3 (*Howard et al., 2019*) as the backbone of YOLOv4.

The remaining parts of the article are arranged as follows: the "Materials and Methods" section offers an introduction to data set acquisition, YOLOv4-tiny, improved YOLOv4-tiny, transfer learning, experimental setup, and evaluation indicators. In the "Results" section we describe the experimental results. In the "Discussion" section we discuss the results related to our application. In the "Conclusion" section we summarize our research and look at future work.

# MATERIALS AND METHODS

## Data sets acquisition

A total of 1,500 images of human skin were collected on the internet, and we considered factors such as position, illumination, skin color, blurring, and the presence of water mist. The position factor meant that the skin could appear in the middle or border of an image. In some images, the skin area was occluded. The difference of illumination in the data sets provided robustness. The data sets included people with fair skin, medium skin, and dark skin color. Our data sets included both clear and blurred pictures. The blur degree needed to ensure that the skin area in a picture was recognizable to the naked eye. In the data sets, some pictures included water mist and some did not include water mist. Some images included all the areas of the human body, and other images only included some areas. Ultimately, 1,000 images were selected based on image quality. We counted the number of pictures using the above factors, and the results are shown in Table 1. According to different regions, we divided skin into six categories: "Face_skin", "Trunk_skin", "Upperlimb_skin", "Lowerlimb_skin", "Hand_skin", and "Foot_skin". The image annotation tool LabelImg (*Bhatt et al., 2022*) was used to generate XML files corresponding to the images. The XML file includes the file name, ground truth information, and category information.

## YOLOv4-tiny

The structure of YOLOv4-tiny is shown in Fig. 1. The backbone is CSPDarknet53-tiny, which is utilized for feature extraction. CSPDarknet53-tiny is composed of

**Table 1 The number of pictures for different factors.**

| Skin position | | Occlusion | | Illumination | | Blur degree | | Water mist | | Number of regional categories | | Skin color | |
|---|---|---|---|---|---|---|---|---|---|---|---|---|---|
| Middle | 589 | Yes | 687 | Sufficient | 708 | Blur | 424 | Yes | 323 | All | 289 | Fair | 277 |
| Border | 411 | No | 313 | Insufficient | 292 | Clear | 576 | No | 677 | Some | 711 | Medium | 421 |
| – | – | – | – | – | – | – | – | – | – | – | – | Dark | 302 |

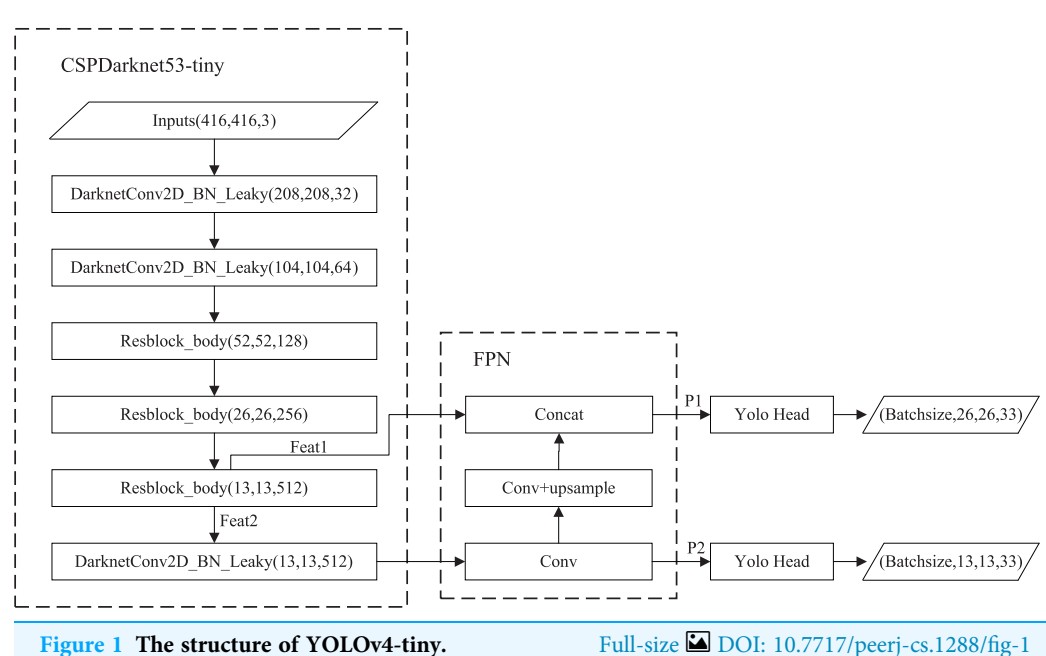

**Figure 1 The structure of YOLOv4-tiny.**

DarknetConv2D_BN_Leaky modules and Resblock_body modules. A DarknetConv2D_BN_Leaky module combines a two-dimensional convolutional layer, normalized processing layer, and activation function. The Mish activation function (*Misra, 2019*) in the YOLOv4 is replaced by a Leaky Relu function (*He et al., 2015*) to improve detection speed. The structure of Resblock_body is illustrated in Fig. 2. The skip connection can better combine semantic information and let the model converge quickly, preventing both model degradation and gradient disappearance (*Furusho & Ikeda, 2020*). Feat1 and Feat2 are the output feature layers from the Resblock_body module. The Feat2 output branch of the first two Resblock_body modules is the input of the next module. FPN (*Lin et al., 2017*) is used to enhance feature extraction and perform feature fusion to combine feature information at different scales. For the output of the third Resblock_body module, Feat1 is directly used as the first input of the FPN. The second input of the FPN is the result obtained by processing Feat2 using the DarknetConv2D_BN_Leaky module. The output P2 of FPN is obtained using convolution processing on the second input of the FPN. The output P1 of FPN is obtained by stacking Feat1 and the result obtained using convolution and up-sampling operations on P2. The structure of FPN is simple, allowing YOLOv4-tiny to have excellent real-time performance. Compared with YOLOv4, YOLOv4-tiny has two detection heads and predicts at two scales. The YOLO head is used to obtain classification

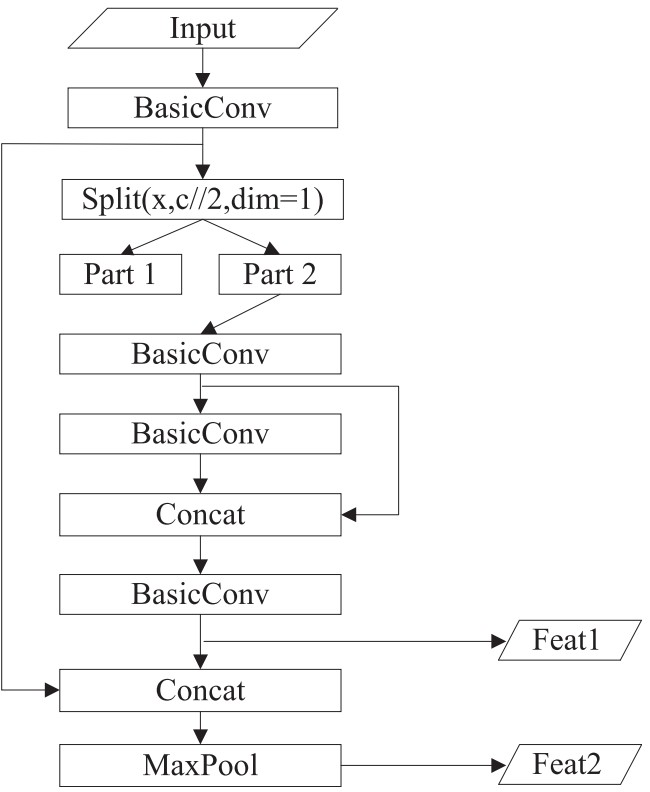

**Figure 2 The structure of Resblock_body.**

and regression prediction results. The structure of the YOLO head is straightforward. The two feature layers for prediction are acquired using a small amount of convolution of P1 and P2. YOLOv4-tiny still makes the detection based on anchors, using fixed-size anchors as a prior for object boxes, tiling many anchors on images, and adjusting anchors to bounding boxes by the prediction results. "13 × 13" and "26 × 26" represent the granularity of grids. "33" represents the prediction results adapted to our application, *i.e.*, 3 × (4 + 1 + 6), where "3" represents the number of anchors, "4" indicates the number of location parameters, "1" denotes the confidence score, and "6" is the number of categories to be identified.

The loss function includes bounding box location loss $L_{loc}$, classification loss $L_{cls}$, and confidence loss $L_{conf}$. The overall loss $L$ is calculated as Eq. (1).

$$L = L_{loc} + L_{cls} + L_{conf} \tag{1}$$

$L_{loc}$ measures the position error between the prediction box and the GT box. The evaluation indicators include IOU, GIOU (*Rezatofighi et al., 2019*), DIOU, and CIOU (*Zheng et al., 2019*), as summarized in Table 2. We introduce CIOU loss as $L_{loc}$, as indicated in Eq. (2).

$$L_{loc} = IoU + \rho^2(b, b^{gt})/d^2 + \alpha\upsilon \tag{2}$$

**Table 2 Summary of IOU, GIOU, DIOU, and CIOU.**

| | Features | Shortcomings |
|---|---|---|
| IOU | Representing the ratio of intersection and union of the GT box and the prediction box | When the prediction box and the GT box do not intersect, the loss function is not differentiable, leading losses cannot propagate |
| GIOU | Scale invariant | Slow convergence speed and low positioning accuracy |
| DIOU | Overlapping area and center point distance are taken into account | Widely used in post-processing |
| CIOU | The consistency of aspect ratio is considered on the basis of DIOU | Widely used in post-processing |

$$v = 4/\left(\pi^2\right) * \left(\arctan\left(w^{gt}/h^{gt}\right) - \arctan\left(w/h\right)\right)^2 \tag{3}$$

$$\alpha = v/v(v + 1 - IoU) \tag{4}$$

where $\rho^2(b, b^{gt})$ represents the European distance between the central points of the prediction box and the GT box, $d$ represents the diagonal distance of the minimum area enclosing the prediction box and the GT box, $\alpha$ is weight, and $v$ expresses the consistency of the aspect ratio. $v$ and $\alpha$ are calculated as demonstrated in Eqs. (3) and (4).

$L_{cls}$ measures the category error between the prediction box and the GT box, as shown in Eq. (5). $K \times K$ represents the number of grids on feature maps of different scales, and $c$ represents the category. If the $j$-th prior box of the $i$-th grid has objects to be predicted, $I_{ij}^{obj} = 1$; otherwise, $I_{ij}^{obj} = 0$. $q_i(c)$ and $p_i(c)$ represent the actual value and predicted value of the probability that the $j$-th prior box of the $i$-th grid belongs to category $c$, respectively. The $L_{cls}$ is optimized using a label smoothing approach to suppress the overfitting problem during training (Zhang et al., 2021). $q_i(c)$ is expressed as Eq. (6) where $y_{true}(c)$ represents the one-hot hard label, $\varepsilon$ is a constant, and $N$ represents the total number of categories.

$$L_{cls} = -\sum_{i=0}^{K\times K} I_{ij}^{obj} \sum_{c\in classes} \left[q_i(c)\log\left(p_i(c)\right) + (1 - q_i(c))\log\left(1 - p_i(c)\right)\right] \tag{5}$$

$$q_i(c) = (1 - \varepsilon)\, y_{true}(c) + \frac{\varepsilon}{N} \tag{6}$$

$L_{conf}$ adopts a cross-entropy loss function, as shown in Eq. (7). $M$ represents the number of prior boxes. $D_i$ and $C_i$ represent the actual and predicted values of confidence. If the $j$-th prior box of the $i$-th grid has no object to be predicted, $I_{ij}^{noobj} = 1$; otherwise, $I_{ij}^{noobj} = 0$.

$$L_{conf} = \sum_{i=0}^{K\times K}\sum_{j=0}^{M} I_{ij}^{obj}\left[D_i\log\left(C_i\right) + (1 - D_i)\log\left(1 - C_i\right)\right]$$

$$- \sum_{i=0}^{K\times K}\sum_{j=0}^{M} I_{ij}^{noobj}\left[D_i\log\left(C_i\right) + (1 - D_i)\log\left(1 - C_i\right)\right] \tag{7}$$

## Improved YOLOv4-tiny based on attention mechanisms

The attention mechanism has a variety of implementations (Niu, Zhong & Yu, 2022). The core of the attention mechanism is to make the network pay attention to needed areas. In general, attention mechanisms can be divided into the channel attention mechanism, the

spatial attention mechanism, and a combination of the two (*Tian et al., 2021*). In this article, the following attention mechanisms were used:

(1) Squeeze-and-excitation (SE) (*Hu, Shen & Sun, 2018*). SE is a typical implementation of the channel attention mechanism that obtains the weights of each channel in the feature maps. The inter-dependencies among channels are modeled explicitly. Instead of introducing a new-built spatial dimension for the fusion of feature channels, SE uses a feature rescaling strategy. Specifically, the importance of each channel is acquired spontaneously by self-learning. SE includes squeeze and excitation operations. The squeeze operation conducts feature compression across the spatial dimension, and converts a two-dimensional feature map into a real number that owns a global receptive field. The output size matches the number of input channels. The excitation operation is equivalent to the mechanics of gates in recurrent neural networks, where weights are created for each channel employing learned parameters, and explicitly models the correlation between feature channels. Finally, the weights, which are output by excitation operations, represent the importance of each channel. The rescaling of features in the channel dimension is accomplished by multiplying the weights by features of each channel (*Huang et al., 2019*). The specific implementation of SE is shown in Fig. 3.

(2) Efficient channel attention (ECA). ECA is an improved version of SE. *Wang et al. (2020)* argued that seizing all channel dependencies is ineffective and unessential for the SE block. Convolution operation owns the cross-channel information capture capability. ECA removes the fully connected layer of SE and learns weights by 1D convolution operation on the globally averaged pooled features. The specific implementation of ECA is shown in Fig. 4.

(3) Convolutional block attention module (CBAM). CBAM (*Woo et al., 2018*) performs channel attention and spatial attention mechanism processing for feature maps, as shown in Fig. 5. The implementation of the channel attention module (CAM) can be divided into two parts. Global average pooling and maximum global pooling are performed separately for the input feature maps. The outputs are processed using a shared, fully connected layer. The two processed results and summed, and the sigmoid operation is taken to obtain the weights of each channel of the input features. The weights are multiplied by the original input features to get the output of CAM. The spatial attention module (SAM) takes the maximum and average value on each channel of each feature point. The two results are stacked. The number of channels should be adjusted using a convolution operation. The weights of each feature point of the input features is determined using the sigmoid function. The output is obtained by multiplying the weights by the original input features.

In this study, the above attention mechanisms are applied to YOLOV4-tiny. As shown in Fig. 6, we added attention mechanisms on the two feature layers extracted from the backbone network and attention mechanisms on the up-sampled results in FPN.

## Transfer learning

Training a network from scratch requires an enormous amount of labeled data. Manual labeling of data sets is time-consuming and labor-intensive, which introduces the possibility of human error. Small data sets combined with transfer learning techniques can

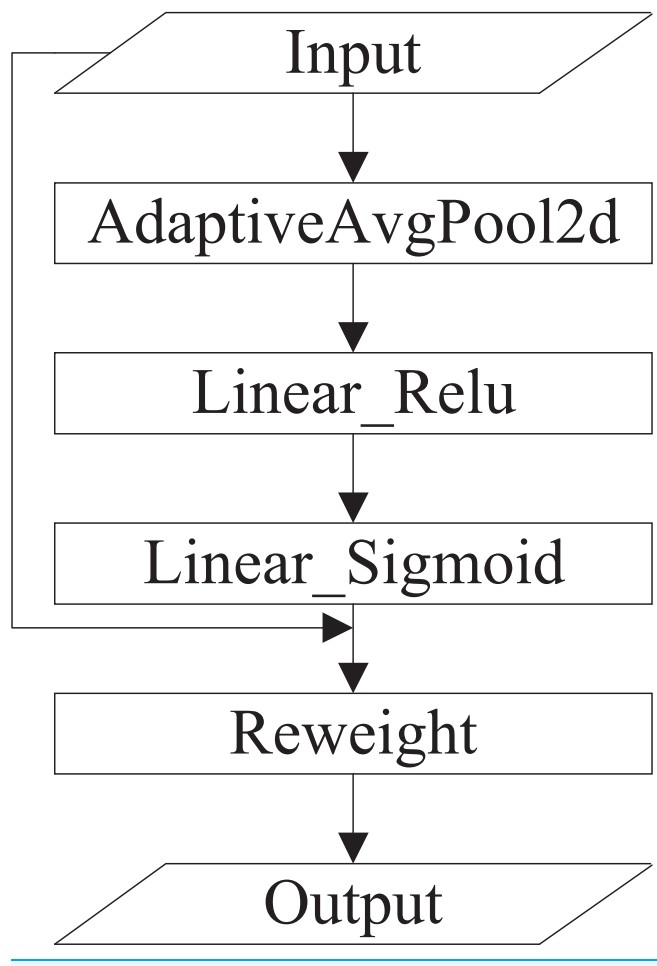

**Figure 3** The specific implementation of SE.

quickly produce a desirable model (*Pratondo & Bramantoro, 2022*). The ImageNet contains more than 14 million images covering more than 20,000 categories, of which more than one million images have explicit annotations and corresponding labels at objects' locations in the image (*Russakovsky et al., 2015*). The pre-trained models on ImageNet can learn fundamental features such as textures and lines, which are general in object detection. All models use the pre-trained weights on the ImageNet as the initial weights in this study.

## Experimental setup and evaluation indicators

For the fairness of model comparison, we used the same data sets as our previous work (*Li et al., 2021*), with a ratio of 60%:20%:20% for the training, validation, and test sets. All models were trained with the help of the high-performance computing center of the University of Shanghai for Science and Technology. Mosaic data augmentation was used in the training process in which four randomly stitched images were input to the network for training to increase the background diversity (*Bin et al., 2022*). In order to compare with

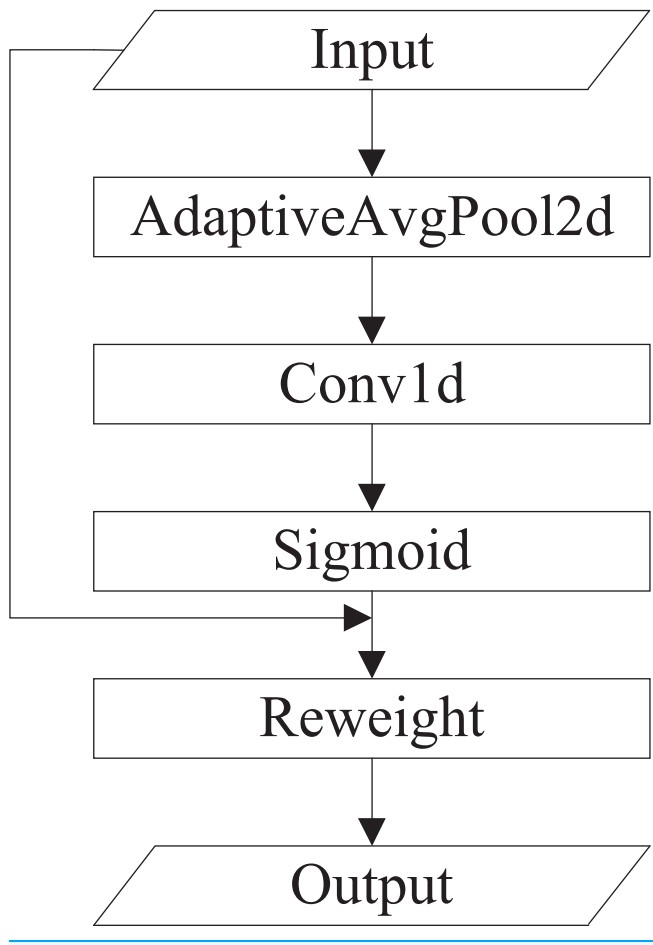

**Figure 4** The specific implementation of ECA.

other lightweight methods, we replaced the backbone of YOLOv4 with MobileNetV1, MobileNetV2, and MobileNetV3 to obtain three network architectures. We used the Pytorch framework for model building and training. The initial value of the learning rate was set to 0.001 and the decay rate was set to 0.01. The batch size was set to 16, which indicated the number of images input to the model for training every time. SGD was utilized as the optimizer for model training. When training, the weights of the backbone were frozen first for 50 epochs, and all weights were trained after 50 epochs, which increased the convergence speed and training performance of models.

Recall and precision can be used to measure performance but are not fully representative of detector quality. Many sets of recall and precision values are obtained by taking different thresholds. First, plot a P-R curve (*Naing et al., 2022*). AP characterized the area enclosed by the P-R curve and the coordinate axes. The sum of the AP values of all classes was then divided using the total number of classes to get mAP, which is the crucial evaluation metric of detectors for multiple category detection. Our research is application oriented. In addition to the mAP indicators, we also focused on the model size. The smaller model and the higher mAP were more desirable for our embedded applications.

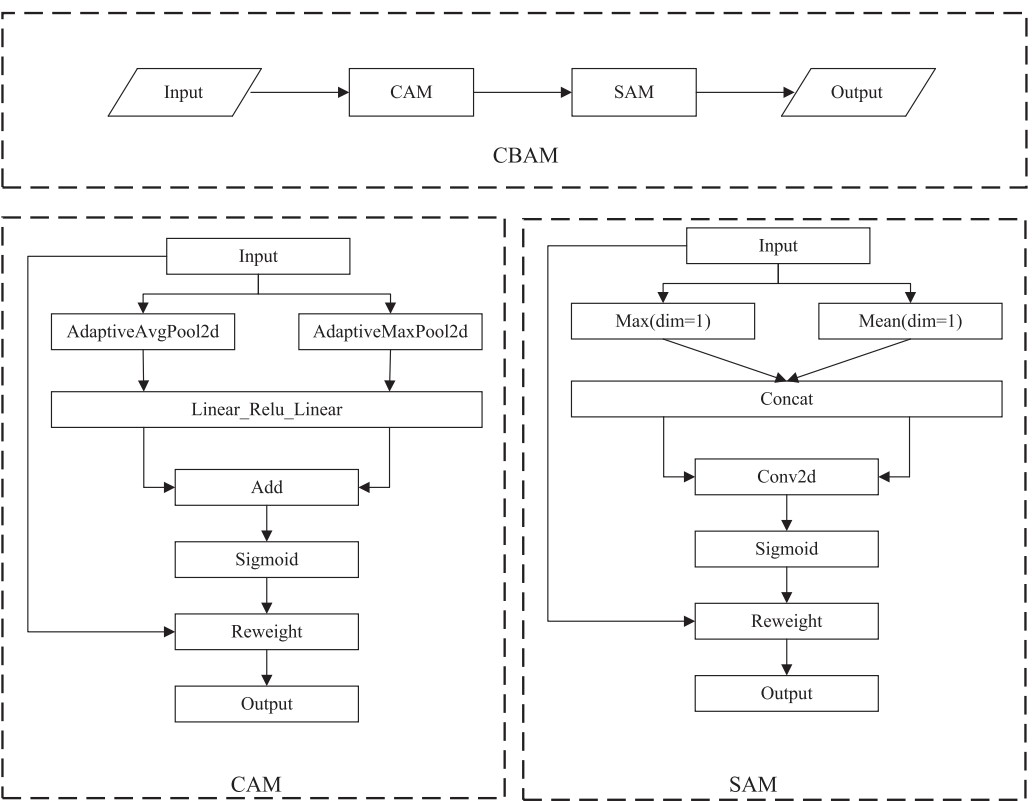

**Figure 5  The specific implementation of CBAM.**

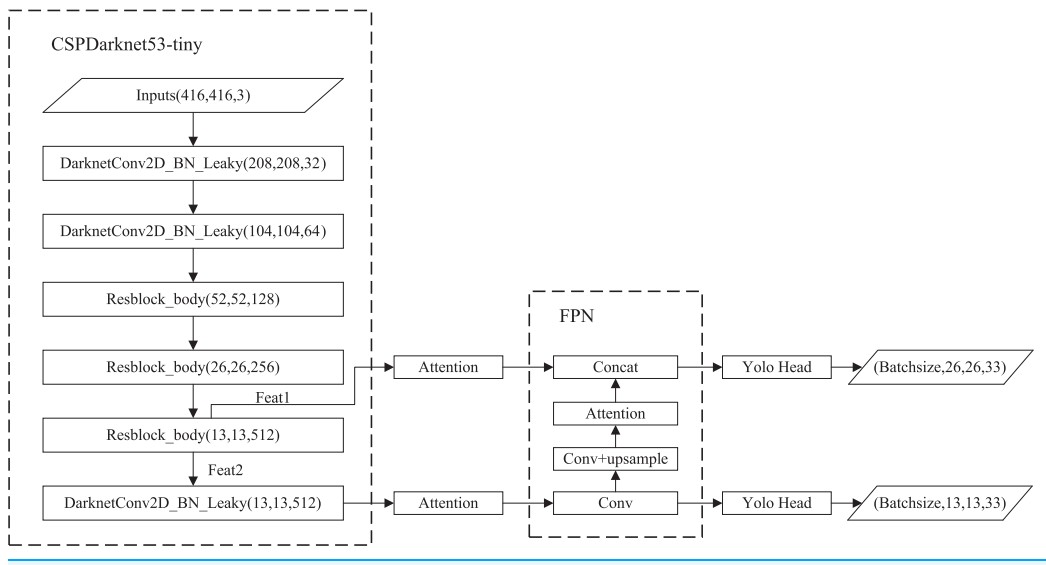

**Figure 6  Improved YOLOv4-tiny based on attention mechanisms.**

**Table 3 Model information.**

| Model | Attention | mAP | Weight file (MB) |
|---|---|---|---|
| YOLOv4-tiny | – | 52.5% | 22.4 |
| SE_YOLOv4-tiny | SE | 51.6% | 22.6 |
| CBAM_YOLOv4-tiny | CBAM | 57.2% | 22.8 |
| ECA_YOLOv4-tiny | ECA | 53.6% | 22.4 |
| MobileNetV1_YOLOv4 | – | 60.5% | 51.1 |
| MobileNetV2_YOLOv4 | – | 60.2% | 46.5 |
| MobileNetV3_YOLOv4 | – | 61.8% | 53.7 |

**Table 4 Model comparison results in previous work.**

| Models | Backbone | mAP |
|---|---|---|
| Faster R-CNN | ResNet50 | 0.72 |
| Faster R-CNN | MobileNetV2 | 0.55 |
| YOLOv3 | DarkNet53 | 0.70 |
| YOLOv4 | CSPDarkNet53 | 0.78 |
| CenterNet | Hourglass | 0.66 |

## RESULTS

After the training was completed, models were selected based on the results of the validation sets and the performance was tested using the test sets. The mAPs and weight file information of models are exhibited in Table 3.

In our previous work, the mAP of YOLOv4 reached 78%, as shown in Table 4. Also, it had a weight file of 244 MB. After the light-weighting process, the mAP of YOLOv4-tiny was 67.3% of YOLOv4, but the weight file was reduced to 9.2% of YOLOv4. Based on YOLOv4-tiny, we added attention mechanisms as shown in Fig. 6. As can be seen in Table 3, mAP is reduced by 0.9% after adding the SE. There was a 1.1% improvement in mAP after adding the ECA. The mAP increased by nearly 5% with the addition of CBAM. After adding ECA, the weight file hardly increased. After adding SE, the weight file increased by 0.2 M. After adding CBAM, the weight file increased by 0.4 M. MobileNetV1_YOLOv4, MobileNetV2_YOLOv4, and MobileNetV3_YOLOv4 represent the model obtained using MobileNetV1, MobileNetV2, and MobileNetV3 as the backbone of YOLOv4. Table 3 shows that the maximum weight file of YOLOv4-tiny series (including YOLOv4-tiny and three improved YOLOv4-tiny based on attention mechanism) was 22.8 MB, and the minimum weight file of MobileNetVX_ YOLOv4 (X = 1, 2, 3) was 46.5 MB. Also, the maximum mAP of YOLOv4-tiny series was 57.2%, and the minimum mAP of MobileNetVX_ YOLOv4 (X = 1, 2, 3) was 60.2%. Overall, the mAP of MobileNetVX_ YOLOv4 (X = 1, 2, 3) was higher than that of YOLOv4-tiny series, but the weight file of MobileNetVX_ YOLOv4 (X = 1, 2, 3) was bigger than that of the YOLOv4-tiny series. The AP values for the six categories are shown in Table 5. CBAM_YOLOv4-tiny achieved the highest AP values for the face and foot, MobileNetV3_ YOLOv4 achieved the highest AP

**Table 5  AP values for the six categories.**

|  | Face_skin | Hand_skin | Upperlimb_skin | Lowerlimb_skin | Trunk_skin | Foot_skin |
|---|---|---|---|---|---|---|
| YOLOv4-tiny | 0.97 | 0.68 | 0.57 | 0.54 | 0.21 | 0.18 |
| SE_YOLOv4-tiny | 0.96 | 0.67 | 0.55 | 0.55 | 0.17 | 0.21 |
| CBAM_YOLOv4-tiny | 0.97 | 0.72 | 0.58 | 0.56 | 0.30 | 0.30 |
| ECA_YOLOv4-tiny | 0.97 | 0.70 | 0.60 | 0.51 | 0.21 | 0.23 |
| MobileNetV1_ YOLOv4 | 0.96 | 0.70 | 0.70 | 0.64 | 0.41 | 0.22 |
| MobileNetV2_ YOLOv4 | 0.96 | 0.71 | 0.73 | 0.67 | 0.40 | 0.14 |
| MobileNetV3_ YOLOv4 | 0.97 | 0.76 | 0.69 | 0.69 | 0.34 | 0.26 |

values for the hand and lower limb, MobileNetV2_ YOLOv4 achieved the highest AP values for the upper limb, and MobileNetV1_ YOLOv4 achieved the highest AP value for the trunk. P-R curves are shown in Fig. 7.

In embedded applications, we believe that model quality was not only related to mAP, but also to model size. We hoped to achieve a better balance between model size and mAP. The size of the weight file could reflect the model size to some extent. Based on the above analysis, a comprehensive indicator $W$ was established to describe the balance, as shown in Eq. (8), where $A = U(i) - U(0)$ and $B = V(i) - V(0)$. $U(0)$ represents the weight file size of the original YOLOv4-tiny, $U(i)$ represents the weight file size of the other model, $V(0)$ represents the mAP of the original YOLOv4-tiny, and $V(i)$ represents the mAP of the other model. The smaller the $A$ is, the smaller the model is. The larger the $B$ is, the better mAP the model has. Intuitively, establishing $W = B/A$ can ensure that the larger the $W$ is, the better the balance between mAP and model size. Considering that $A \geq 0$ ($U(i) \geq U(0)$ seen from Table 3) and $y = e^x$ is a monotonically increasing function, we finally used $e^A$ instead of $A$, which can avoid the situation where the denominator equals 0 and can ensure that $W$ decreases with the increase of $A$. Based on the above information, $W$ can depict the balance between mAP and model size. After calculation, $A$, $B$, and $W$ are indicated in Table 6. $W$ of CBAM_YOLOv4-tiny is highest and CBAM_YOLOv4-tiny achieved the best balance.

$$W = \frac{B}{e^A} (A \geq 0) \tag{8}$$

## DISCUSSION

To perform the bathing tasks, we needed to recognize the area that needed to be bathed in the bathing scenario and send the recognition information to the bathing robot arm for bathing behavior planning, as shown in Fig. 8. By combining the skin detection results of 2D images with the depth information obtained from the depth camera, we could model the localization of targets in 3D space. In order to facilitate the robot to implement distinct bathing patterns for areas of the body, we needed to identify the skin located at diverse parts of the body. Therefore, we built small data sets in the bathing scenarios to be used as learning samples for object detection models. The manual annotation was performed with the labelImg tool to classify skin regions into six categories according to different parts.

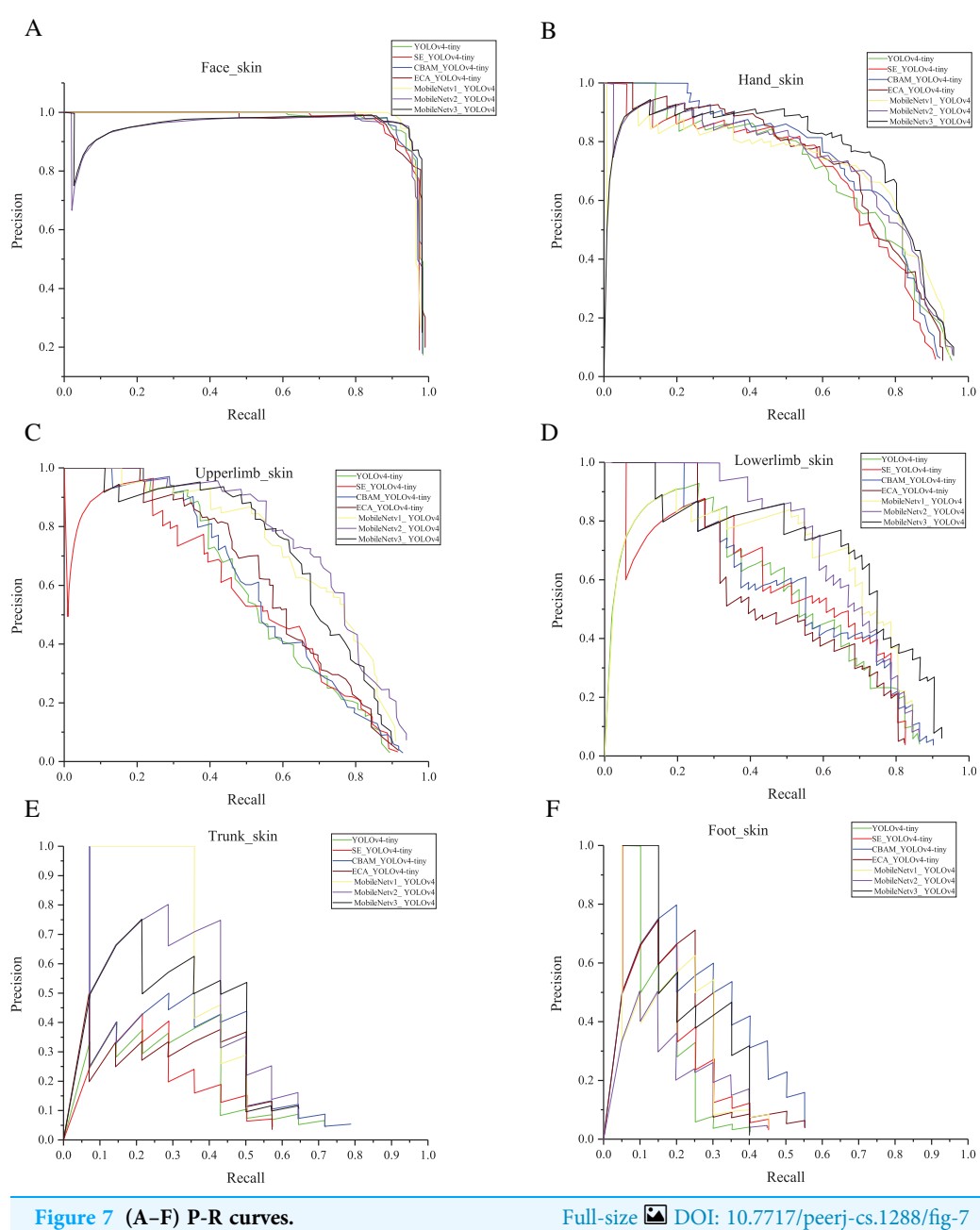

**Figure 7 (A–F) P-R curves.**

**Table 6 A, B, and W of all models.**

| Models | A | B | W |
|---|---|---|---|
| SE_YOLOv4-tiny | 0.2 | −0.9 | −0.74 |
| CBAM_YOLOv4-tiny | 0.4 | 4.7 | 3.15 |
| ECA_YOLOv4-tiny | 0 | 1.1 | 1.1 |
| MobileNetV1_YOLOv4 | 28.7 | 8 | 2.75 |
| MobileNetV2_YOLOv4 | 24.1 | 7.7 | 2.63 |
| MobileNetV3_YOLOv4 | 31.3 | 9.3 | 2.37 |

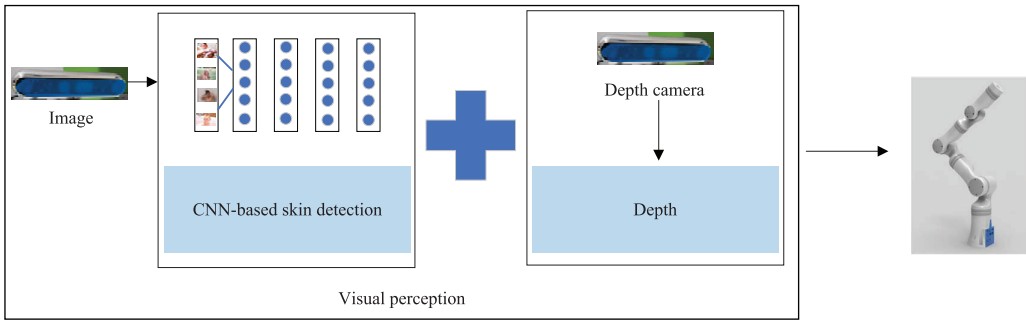

**Figure 8** The perception process in the bathing tasks: achieving the three-dimensional positioning of the target.

Among the object detection algorithms, one-stage detection algorithms are faster than two-stage and are suitable for application in our scenario where real-time performance is required. In our previous work, we explored the effectiveness of object detection models for skin detection with multiple classifications and found the best YOLOv4 model from five models. For easy deployment, we used lightweight YOLOv4-tiny and imposed three kinds of attention mechanisms on YOLOv4-tiny. We found that both CBAM and ECA improved the detection effect, yet SE made the detection effect worse instead, which implies that we need to carefully choose the attention mechanism during practice. We also used another lightweight method that replaced the backbone of YOLOv4 with MobileNetV1, MobileNetV2, and MobileNetV3. They obtained a higher mAP than YOLOv4-tiny series and were accompanied by a larger weight file.

Compared with Salah's work, our data sets included images with six types of labels for network training instead of skin and non-skin patches. We were not only able to discern between skin and non-skin patches, but also the body part. To the best of our knowledge, this is the first time a study on skin detection that has been able to identify different body parts. The innovation of this article is in the use of CNN-based object detection algorithms to complete regional skin detection for bathing tasks with embedded applications.

When developing assistive robots, visual information is frequently used, especially since there is are a number of image processing algorithms. Our application scenarios are special and, fortunately, we also considered privacy protection. Mitigating damage to privacy mainly starts when: (1) the data are not stored locally and the whole system does not display RGB information (our hardware computing platform uses jetson TX2, which has limited storage space and does not support the storage of visual data in the bath scene); (2) the whole system is not connected to the network, ensuring that the data do not have the risk of transmission; (3) the data are processed on TX2 rather than transmitted to the cloud for processing. In fact, we hoped to find the two-dimensional pixel coordinate information with the help of a mature algorithm using RGB and combine the depth information of the depth camera to model object regions in three-dimensional space. In addition, we mainly applied the bathing robot to semi-disabled elderly people who generally hope to complete bathing independently of nursing staff. The bathing robot can provide the elderly with the opportunity to take care of themselves while bathing.

Compared with relying on nursing staff to complete bathing tasks, using the bathing robot can maintain the dignity of the elderly to the greatest extent.

There has been little research on object detection-based skin detection combined with robotic arms for bathing tasks. Our study explored the YOLOv4-tiny and which attention mechanism works best on YOLOv4-tiny. However, the YOLOv4-tiny showed a reduction in mAP compared with the YOLOv4, creating some challenges for high detection accuracy (*Zhao et al., 2022b*). The relatively small number of trunks in the data sets resulted in the poor detection of trunks. The foot occupies a small area in the whole-body range. Foot features tend to disappear with repeated down-sampling operations, resulting in poor detection of the foot.

## CONCLUSION

When using robots for autonomous bathing tasks, skin detection needs to be accomplished first. To facilitate the embedded deployment, we used YOLOv4-tiny, a lightweight model of YOLOv4, for skin detection based on our previous work. Three kinds of attention mechanisms were overlaid in the YOLOv4-tiny, and we used the test sets to test the performance of models. Compared to the original YOLOv4-tiny, the YOLOv4-tiny combined with the CBAM or ECA attention modules showed a certain increase in mAP, while the addition of SE produced some degree of decrease. It is feasible to use attention mechanisms for performance improvement of YOLOv4-tiny, but not every attention mechanism is suitable. Compared with the lightweight method of using MobileNetV1/MobileNetV2/MobileNetV3 as the backbone of YOLOv4, the method of using YOLOv4-tiny and CBAM achieves a better balance between model size and detection effect. In future work, we will improve the detection for trunk and foot by expanding the trunk and foot samples in the self-built data sets, and aim to guarantee deployment performance while achieving high detection accuracy. Then, we will convert the best model into an open neural network exchange model for easy deployment.

## ACKNOWLEDGEMENTS

We want to express our gratitude to the high-performance computing center of the University of Shanghai for Science and Technology.

### Funding

This work was supported by the National Natural Science Foundation of China (No. 62073224). The funders had no role in study design, data collection and analysis, decision to publish, or preparation of the manuscript.

### Grant Disclosures

The following grant information was disclosed by the authors:
National Natural Science Foundation of China: 62073224.

## Competing Interests

The authors declare that they have no competing interests.

## Author Contributions

- Ping Li conceived and designed the experiments, performed the experiments, analyzed the data, performed the computation work, prepared figures and/or tables, and approved the final draft.
- Taiyu Han performed the experiments, authored or reviewed drafts of the article, and approved the final draft.
- Yifei Ren analyzed the data, authored or reviewed drafts of the article, and approved the final draft.
- Peng Xu performed the computation work, prepared figures and/or tables, and approved the final draft.
- Hongliu Yu performed the computation work, prepared figures and/or tables, and approved the final draft.

## Data Availability

The code is available at GitHub https://github.com/liping22/YOLOv4-tiny-attention-Pytorch/releases/tag/0.2; liping22. (2022). liping22/YOLOv4-tiny-attention-Pytorch: YOLOv4-tiny-attention-Pytorch 0.2 (0.2). Zenodo. https://doi.org/10.5281/zenodo.7152767.

The data is available at FigShare: Li, Ping (2022): Data sets.rar. figshare. Dataset. https://doi.org/10.6084/m9.figshare.21282396.v1.

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
