# Peer review of "Improved YOLOv4-tiny based on attention mechanism for skin detection"

_PeerJ Computer Science, doi:10.7717/peerj-cs.1288_

## Round 0.1 · original submission · Major Revisions

Please incorporate the reviewers' comments.

Reviewer 1 ·

Basic reporting

This paper proposed a skin detection method based on attention mechanism to improve YOLOv4-tiny. The automatic bathing robot automatically changes the bathing mode by detecting different skin areas of the human body in the bathing scene through a visual sensor. This paper enhanced the feature extraction and feature fusion capabilities of the network by adding three attention mechanisms. And ultimately it improved the detection effect of different skin regions of the human body.
The subject is of interest. The following problems should be addressed properly or explained reasonably.
In introduction, the focus of related works is not clear. Many traditional methods are introduced. In fact, this work is focus on the methods based on deep learning theory. Current deep learning related SOTA methods should be mainly described.

Experimental design

In the experimental setting, the comparison method only uses YOLOv4-tiny. It is not convincing enough to say that the YOLOv4 algorithm has a high mAP in skin detection in bathing scenes. Other recent models should be compared to the YOLOv4-tiny model and the results should be clearly represented.
In the Materials & Methods, it contains not only the research content of other scholars (three attention mechanisms and the network structure of YOLOv4), but also the experimental data of some methods and the experimental content of this paper (improved structure of YOLOv4-tiny). The authors need to highlight this paper's innovative contributions.

Validity of the findings

The data is a self-made dataset, a total of 1500 images containing human skin are collected, considering factors such as location, illumination, resolution, blur and water mist. The total amount of data is small, and the data samples should be clearly explained.
This paper studied the skin detection technology that can identify different parts of the human body for the first time and made a data set. However, the visual sensor to detect the skin involves personal privacy issues. The author should explain the problem clearly to better demonstrate the advantages of the proposed method.

Additional comments

In the experimental results, the establishment of comprehensive evaluation index W. A represents the change of weight file, and B represents the change of mAP. The basis for the establishment of comprehensive evaluation index should be introduced in detail. The source of the index relationship between A and B, please analysis.
The authors are suggested to proofread the draft, including formula letters, labels, picture size, etc. The method abbreviations in the text, tables and pictures are not uniform, such as CBAM_YOLOv4-tiny in the text and tables, and YOLOv4-tiny_CBAM in the pictures. Pictures and tables are located at the end of the article, typesetting unreasonable. English writing should be improved.

Reviewer 2 ·

Basic reporting

The main idea of this manuscript sounds good. Not only recognizes the skin but also the area. The writing is well-presented and easy to understand.

Experimental design

The method used is commonly used in computer vision. There is no specific novelty, but still OK because the proposed research tends to be applied. Unfortunately, the equations look messy and poorly presented (Eq. 5 and 6). The equation is not just an ornament in the manuscript but explains the method you are using and where your method stands.

Validity of the findings

The number of datasets may still be increased for further research. For now it might be enough. Unfortunately, I don't see a detailed description of this data set. The author mentions in line 100 factors that they use in the dataset. The details of the data set for each factor are not well explained. Does each image contain all the areas to be recognized? Are there areas covered by body members? How many images of each factor? One factor may produce high accuracy but not the other factors. Here the authors can adjust the number of data sets such that accuracy looks good. Furthermore, the authors did not take skin colour into account. Ideally, the proposed model is capable of handling a wide range of skin tones, regardless of ethnicity. The authors did not show the robustness of the proposed model against the skin of various races.

---

## Round 0.2 · Minor Revisions

Address the remaining issues within the manuscript.

Reviewer 2 ·

Basic reporting

The paper has been significantly improved.

Experimental design

I have only a few notes for this revised version of the manuscript. I hope that in the final version, the author uses the notation that is already commonly used, for example, flowcharts. Inputs and outputs on the flow chart are different from the process. There are many kinds of symbols, but at least the writer needs to differentiate the block diagram for input/output and process in Figure 5 to increase the readability of the diagram.

Validity of the findings

I am satisfied with the improvements made by the author

Additional comments

-

---

## Round 0.3 · accepted · Accept

This manuscript is ready for publication.

Reviewer 2 ·

Basic reporting

no issue

Experimental design

no issue

Validity of the findings

no issue

Additional comments

I am satisfied with the latest revision.